# Qualitative Exploration of Anesthesia Providers’ Perceptions Regarding Philips Visual Patient Avatar in Clinical Practice

**DOI:** 10.3390/bioengineering11040323

**Published:** 2024-03-27

**Authors:** Cynthia A. Hunn, Justyna Lunkiewicz, Christoph B. Noethiger, David W. Tscholl, Greta Gasciauskaite

**Affiliations:** Institute of Anesthesiology, University and University Hospital Zurich, Raemistrasse 100, 8091 Zurich, Switzerland

**Keywords:** patient monitoring, qualitative research, situation awareness, user-centered design, Visual Patient Avatar

## Abstract

The Philips Visual Patient Avatar, a user-centered visualization technology, offers an alternative approach to patient monitoring. Computer-based simulation studies indicate that it increases diagnostic accuracy and confidence, while reducing perceived workload. About three months after the technology’s integration into clinical practice, we conducted an assessment among anesthesia providers to determine their views on its strengths, limitations, and overall perceptions. This single-center qualitative study at the University Hospital of Zurich examined anesthesia providers’ perceptions of the Philips Visual Patient Avatar after its implementation. The study included an online survey to identify medical personnel’s opinions on the technology’s strengths and areas for improvement, which were analyzed using thematic analysis. A total of 63 of the 377 invited anesthesia providers (16.7%) responded to the survey. Overall, 163 comments were collected. The most prevalent positive themes were good presentation of specific parameters (16/163; 9.8%) and quick overview/rapid identification of problems (15/163; 9.2%). The most common perceived area for improvement was the ability to adjust the visualization thresholds of Visual Patient Avatar, which represent the physiological upper and lower vital-sign limits (33/163; 20.3%). The study showed that users consider Philips Visual Patient Avatar a valuable asset in anesthesia, allowing for easier identification of underlying problems. However, the study also revealed a user desire for the ability to freely adjust the thresholds of the Visual Patient Avatar by the handling caregivers, which were fixed to the departmental standard during the study.

## 1. Introduction

Technological advances have significantly improved patient care [1,2], particularly regarding patient monitoring in the perioperative [3] and critical care settings [4,5]. As the number of parameters recorded by patient monitoring devices increases, the growing volume of perceived data requires more complex interpretation from clinicians and increases their cognitive load [6,7,8]. This contributes to insufficient situation awareness among healthcare providers, a factor that can be attributed to nearly 80% of treatment errors in the perioperative setting [9,10]. Therefore, it is imperative to help healthcare professionals manage the overwhelming amount of information, while increasing their situation awareness and understanding of the patient’s health status.

Based on cognitive psychology and neuroscience principles [11,12,13], humans can perceive, comprehend, and process a user-centered, integrative visual language more effectively than a technology-centered approach that measures specific parameters and displays them as separate numbers and waves [14]. Several tools, such as AlertWatch^®^ OR (AlertWatch, Inc., Ann Arbor, MI, USA) [15], Dynamic Lung Panel (Hamilton Medical AG, Bonaduz, Switzerland) [16], PulmoSight (Mindray Medical International Limited, Shenzhen, China) [17], and HemoSight^TM^ (Mindray Medical International Limited, Shenzhen, China) [18], aim to use this understanding by integrating information in a simplified visual presentation. Another technology that aligns with these principles is Philips Visual Patient Avatar—a user-centered visualization technology designed with a focus on enhancing situation awareness in patient monitoring [19]. It employs colors, shapes, and animations in the form of an avatar to convey vital-sign information (Figure 1) [20]. Several studies have shown the beneficial effects of the situation awareness-enhancing technology by enabling clinicians to better understand and communicate underlying medical issues compared to traditional monitoring methods [20,21,22]. Furthermore, the Visual Patient Avatar has been proven to enhance diagnostic confidence and reduce perceived workload [20,22]. The benefits of this novel visualization technique can be attributed to the design of the technology, which combines different parameters from multiple sensors into a single indicator, enabling healthcare providers to assess the full spectrum of vital signs simultaneously. User-centered design principles support the use of direct visual representations of data to enhance situation awareness [23]. According to Wittgenstein’s *Tractatus Logico-Philosophicus* [24], a logical image has meaningful commonalities with the reality it is intended to represent, as is demonstrated in Philips Visual Patient Avatar. The NASA publication *On Organization of Information: Approach and Early Work* by Degani et al. [25] emphasizes that the highest level of “order and wholeness” can be achieved by consolidating all necessary data into a single display. This approach allows care providers to quickly assess whether all parameters are within the normal range, providing immediate reassurance and reducing the cognitive load associated with examining each number individually. A detailed comparative list of vital parameters presented by conventional monitoring and their respective visualized counterparts by the Visual Patient Avatar can be found in Appendix B (Table A1).

While the functionality and user perceptions of the Visual Patient Avatar have been extensively investigated in various computer-based and high-fidelity simulation studies [20,21,22], the recent introduction of this technology into clinical settings has prompted the need to investigate its real-world application. This study makes a significant contribution by being the first to investigate user perceptions of the Visual Patient Avatar technology in adult anesthesia within real clinical settings.

The objective of this study was to collect and analyze perspectives of anesthesia professionals regarding their perceived benefits and areas for improvement for the implementation of the Philips Visual Patient Avatar in their clinical practice.

## 2. Materials and Methods

### 2.1. Approval and Consent

The study protocol was reviewed by the Cantonal Ethics Committee of the Canton of Zurich, Switzerland, which issued a declaration of non-jurisdiction (Business Management System for Ethics Committees number Req-2021-00756). Each participant implicitly consented to the use of his/her data for research purposes by completing the questionnaire as described in the study invitation. Participation was voluntary and without compensation.

### 2.2. Implementation of Philips Visual Patient Avatar in the Study Center

The Philips Visual Patient Avatar was implemented in March 2023 in the operating rooms of the University Hospital Zurich, where it was initially developed. This pioneering move by the study center marks the first application of the technology in clinical practice. In the early stages of its deployment, educational lectures were held to introduce the concept of the Visual Patient Avatar and to provide training sessions to familiarize clinicians with its use. In addition, an instructional video and a concise reference guide were posted on the study center intranet for reference (Appendix A). The Philips Visual Patient Avatar is regularly used as a split-screen feature, providing a simultaneous view of both conventional patient monitoring and the Visual Patient Avatar (Figure 1).

### 2.3. Study Design

We conducted a researcher-initiated, single-center, qualitative descriptive study to explore anesthesia providers’ perceptions of the Philips Visual Patient Avatar three months after its implementation in the operating rooms of the study center. The study included an online survey with open-ended and closed-ended questions sent to staff anesthesiologists, residents, and nurse anesthetists. The study was conducted at the Institute of Anesthesiology, University Hospital Zurich, Switzerland, during two consecutive weeks in June 2023.

### 2.4. Online Questionnaire

We created an online survey using Google Forms (Alphabet Inc., Mountain View, CA, USA) and distributed it via email to all anesthesia care providers at the study center, including staff anesthesiologists, residents, and nurses. The initial questionnaire was sent on 12 June 2023, and a reminder was sent to the same participants one week later. Data collection was completed on 26 June 2023.

In the survey invitation, we informed potential respondents that this was a scientific project, that the survey would take approximately 3 min to complete, and that their participation was completely voluntary. The translated survey invitation is included in Appendix A.

The survey was divided into two sections (Appendix A). Its development and validation process involved a number of key steps. Initially, we defined clear research objectives to identify experts’ opinions on the positive aspects and areas for improvement of the Philips Visual Patient Avatar in the clinical setting. The first section of the questionnaire was based on these objectives and consisted of open-ended questions that were intentionally broad in order to encourage participants to freely share their experiences in the form of written comments. The second section consisted of five closed-ended questions designed to collect demographic data from the participants—these were nonspecific to ensure the anonymity of the participants. This information included age, gender, professional role (staff anesthesiologist, resident, and nurse anesthetist), the number of years they had been practicing anesthesia, and whether the respondents were familiar with the provided learning material.

Anesthesia team members that were uninvolved with the study provided input to refine the questionnaire and assess its content validity. These colleagues also assisted in identifying issues related to question wording, comprehension, and response options. Revisions were implemented based on their feedback.

### 2.5. Accounting for Bias

We employed a comprehensive study design aimed at preemptively addressing various potential biases. To counteract the likelihood of non-response bias, for example, due to an overly complex or lengthy survey design [26], we constructed the questionnaire to be clear and concise. To reduce the impact of self-selection bias [27], we delivered institute-wide presentations on the Philips Visual Patient Avatar that were supported by scientifically validated simulation studies to ensure that all clinicians had equal exposure to the technology. To minimize the influence of information bias, such as due to an instrument favoring wrong data entry [28], the survey instrument was carefully constructed according to the structured approach described above to ensure its validity and reliability.

### 2.6. Reflexivity

Reflexivity embodies a dynamic continuum of collaborative processes in which researchers conscientiously assess their subjectivity and the contextual influences that shape their research endeavors [29]. With this premise in mind, we aim to describe various forms of reflexivity relevant to our current study.

Personal reflexivity plays a role in the context of our study in terms of the active involvement of the investigators in the development of the Visual Patient Avatar technology. Their motivation lies in evaluating user feedback and addressing concerns, which are the focus of this study.

The main challenge of interpersonal reflexivity stems from the participants’ affiliation with the institution where technologies like Visual Patient and other user-centered systems were conceived, potentially shaping their perspectives.

Methodological reflexivity guided the selection of a qualitative approach that allowed for a comprehensive analysis of user perceptions after technology implementation, deeming this study one of the first in an effort to assess real-world user feedback. Future research could explore alternative methodologies, such as mixed methods, to gain further insights.

Contextual reflexivity highlights the increasing complexity of patient monitoring systems, which require robust support for care providers amidst a data overload. A viable solution may include tailored visualization techniques that present vital-signs data with user needs in mind.

### 2.7. Data Analysis

#### 2.7.1. Open-Ended Questions

We used DeepL (DeepL GmbH, Cologne, Germany. https://www.deepl.com/en/translator, accessed on 13 July 2023), an online translator, to render the responses from German to English. The translated responses can be found in Appendix A.

To analyze the open-ended questions, we followed a six-phase thematic analysis approach to identify the prevailing themes within each open-ended question [30]. After completing the data collection and conducting a comprehensive review of the gathered information, the members of the research team assembled to discuss their overall impressions and the potential themes derived from the collected responses. Following the collaborative discussion, a coding template was developed (Figure 2). Initially, two team members, GG and JL, independently examined the participants’ responses and coded them using the coding template. If any discrepancies arose between the examiners after data coding, a final decision was made in a joint discussion. As recommended for qualitative research, we conducted an analysis to evaluate interrater reliability [31].

#### 2.7.2. Demographic Data

Demographic data were analyzed using Microsoft Excel (Microsoft 365, version 2307, Microsoft Corp., Redmond, WA, USA). We presented demographic data as numerical values and their respective percentage distributions, or medians with interquartile ranges.

## 3. Results

### 3.1. Participant Characteristics

Of the 377 anesthesia care providers contacted, 63 (16.7%) completed the questionnaire. The distribution of staff anesthesiologists, residents, and nurses was approximately equal in number. The majority of participants (90.5%) consulted the training materials provided. Table 1 shows detailed information about the study’s participants.

### 3.2. Statements about Philips Visual Patient Avatar

A total of 163 statements were collected during the survey. Out of these, 51 (31.3%) were categorized as describing positive features of the technology, while 88 (54.0%) were categorized as referring to areas for improvement. In total, 24 (14.7%) of the 163 comments were not assigned to any group and were described as being not codable. The interrater reliability of the initial coding was 85%. After the initial coding, joint discussions were held to address any discrepancies between the examiners, resulting in 100% agreement across all statements. Figure 2 displays the distribution of comments describing positive features and areas for improvement, along with the themes identified, as well as the number and percentage of statements.

#### 3.2.1. Perceived Positive Features of Philips Visual Patient Avatar from Anesthesia Providers’ Perspective

Three main themes were identified to summarize the perceived positive attributes of the Philips Visual Patient Avatar in an anesthesia setting: favorable usability characteristics, appealing graphic design, and intuitiveness.

##### Favorable Usability Characteristics

Among the 163 comments, 24 (14.6%) emphasized this particular issue in the online survey. Participants indicated that the Philips Visual Patient Avatar provides them with a quick overview of a clinical situation, allowing them to rapidly identify underlying problems, as reported in 15 out of the 163 comments (9.2%).


*Visual Patient gives me the ability to quickly identify all problems in a dynamic situation.*
(Participant 16)


*Visual Patient provides a quick overview, especially when things get hectic, without triggering alarms.*
(Participant 24)

Three statements out of the 163 (1.8%) highlighted that the technology offers good visibility of vital parameters from a distance.


*Good visibility of alarms, even from a greater distance.*
(Participant 9)


*Pathologies can be easily detected from a distance.*
(Participant 35)

Three of the 163 comments (1.8%) specifically stated that the Philips Visual Patient Avatar is a valuable additional support that is especially beneficial for beginners compared to the traditional display.


*A tool that can be helpful, especially for beginners.*
(Participant 45)


*A good addition to the numerical display.*
(Participant 19)

##### Appealing Graphic Design

Among the 163 comments, 20 (12.3%) specifically addressed positive features of the Visual Patient Avatars design in the online survey. In addition, 16 comments (9.8%) specifically indicated a preference for the display of certain vital signs.


*I notice when Visual Patient shows a low temperature or poor saturation because the visual appearance changes significantly.*
(Participant 27)


*I like the easier identification of ST changes.*
(Participant 33)

##### Intuitiveness

Among all comments, seven (4.3%) emphasized the intuitiveness of the technology, particularly regarding its user-friendly and easily learnable nature within a clinical context.


*I like the intuitive understanding of my patient’s circulation.*
(Participant 32)


*The visual representation is intuitive and easy to understand.*
(Participant 63)

#### 3.2.2. Perceived Areas of Improvement of Philips Visual Patient Avatar from Anesthesia Providers’ Perspective

Similar to the section on positive features, when discussing perceived areas of improvement, graphic design and usability remained the predominant focal points. Additionally, a noticeable theme that emerged was the issue of precision in information presentation.

##### Identified Areas for Usability Enhancement

This theme received the most comments in the survey, with 55 out of 163 comments (33.7%) expressing opinions on it. Specifically, the most frequently discussed sub-theme was the non-adjustability of alarm thresholds by the handling caregivers, as they were set and fixed based on departmental standards during the study. This sub-theme was mentioned in 33 statements (20.3%).


*Ability to independently adjust limits would be great, so that values that are too high or too low could be adjusted for the individual patient.*
(Participant 13)


*Alarm thresholds need to be adjustable. For example, in thoracic surgery, a “blue” patient with SpO2 92% is often more confusing than helpful.*
(Participant 63)

Among all comments, four (2.5%) expressed that the interpretation of the Philips Visual Patient Avatar becomes more complex when multiple parameters are outside of the normal range.


*If multiple parameters deviate near the limits, the overview is lost because too many stimuli are displayed at the same time.*
(Participant 30)


*When displaying multiple parameters (in ICU patients, neurosurgery, cardiac surgery), it becomes bothersome.*
(Participant 1)

##### Potential Avenues for Enhancing Graphic Design

This theme received 22 out of 163 comments (13.5%) reflecting opinions on it. The dominant sub-theme in this section was improving the presentation of specific parameters, and this sub-theme was mentioned in 12 comments (7.4%).


*The current visual representation during tachycardia is too hectic. I cannot work with the VP for tachycardic patients. It is like a stroboscope.*
(Participant 32)


*The eyes are spooky.*
(Participant 57)

Additionally, space allocation on the split screen was referenced in six statements (3.7%).


*It takes up too much space on the monitor.*
(Participant 5)


*A larger portion of the split monitor should be dedicated to curves and measurements that are incorporated into the avatar.*
(Participant 7)

##### Lack of Precision in Presenting Information

In total, 11 of the 163 comments (6.8%) pointed out an issue with the precision of the information display when working with the Philips Visual Patient Avatar.


*I would like to see more gradations, for example, in cases of hyper- or hypotension or SpO2 (not just good or bad or more than 2 colors).*
(Participant 17)


*Integration of trends (e.g., temperature).*
(Participant 10)

Twenty-four (14.7%) comments were deemed non-codable.

E.g., *The monitors often have problems.*
(Participant 20)

Table 2 provides the main topics identified in the responses to the open-ended questions with the number of statements, percentages, and examples.

## 4. Discussion

This researcher-initiated, single-center, qualitative, descriptive study investigated anesthesia providers’ perceptions of the Philips Visual Patient Avatar three months after its real-world introduction into clinical routine at the University Hospital Zurich. It allowed for the identification of the positive aspects of the technology and highlighted areas for further improvement as perceived by anesthesia providers. The principal findings of the study indicate that participants perceive the Philips Visual Patient Avatar as a valuable and intuitive supplementary tool to the conventional display by providing a good presentation of specific parameters, while revealing a user desire for adjustable thresholds and further refinements in design and information presentation (Figure 3).

Participants stated that the Visual Patient Avatar offers a quick overview of the clinical situation and enables the rapid identification of underlying issues. These findings align with previous computer-based and high-fidelity simulation studies of the user-centered, situation awareness-based technology, whereby participants praised the global overview and intuitive design provided by the Visual Patient Avatar [20,32]. Earlier simulation studies have also demonstrated that integrating numerical and waveform data into an intuitive visualization format, such as the Visual Patient Avatar, can accelerate decision-making, improve clinical diagnosis accuracy, reduce perceived workload, and increase perceived diagnostic confidence [20,22]. Moving forward, these positive outcomes may translate into tangible benefits in the real-world application of the Visual Patient Avatar by addressing the various external factors identified by Endsley et al. that adversely affect a caregiver’s situation awareness [33]. Given that a lack of situation awareness is a significant contributor to healthcare errors, the technology specifically addresses Level 1 situation awareness by helping users to perceive their environment more effectively, which is often the primary shortfall in situation awareness [10]. By improving caregivers’ perception, the technology consequently enhances the subsequent stages of comprehension (Level 2), projection (Level 3), decision-making, and action, thereby improving patient safety [33]. This, in turn, contributes to better clinical practice and enhanced patient care.

In a previous effort to refine the design of the Visual Patient Avatar prior to its introduction into the clinical setting, users were asked to provide feedback on their perceptions of an earlier version of the technology [34]. Compared to the user perceptions of the Visual Patient Avatar prior to its redesign, this study revealed that the updated version of the technology evoked positive feedback from numerous participants concerning the presentation of specific parameters. With this in mind, we recognize the importance of user feedback in optimizing the Visual Patient Avatar and aim to incorporate the ideas and suggestions of anesthesia professionals in future developments.

In this study, a common user desire identified relates to the flexibility of the Visual Patient Avatar thresholds. Approximately one-fifth of the comments suggested that providing adjustable modifications to these thresholds could enhance the practice of anesthesia providers. While maintaining consistent Visual Patient Avatar thresholds offers a standardized reference for professionals according to institutional guidelines defined by an expert group, some participants noted that static thresholds might result in unnecessary visual alerts when acoustic alarm settings differ from the visual indication.

For the duration of the study, the implementation of the Visual Patient Avatar included three predefined anesthesia profiles, each with distinct acoustic alarm limits and fixed Visual Patient thresholds for neonatal, pediatric, and adult patients. While users had the flexibility to adjust acoustic alarm limits, the Visual Patient Avatar thresholds remained fixed to the predefined profile setting. Further research is warranted to explore potential alternatives to the Visual Patient Avatar configuration with fixed thresholds, such as evaluating the effects of user-customizable or acoustic alarm-aligned thresholds.

Alongside the perceived positive features attributed to the presentation of specific parameters, users also provided valuable input on their perceived areas of improvement for the same subtopic. For instance, suggestions for additional visualizations, such as the presentation of a pacemaker function (Participant 38), might guide the development of Visual Patient Avatar to present a wider range of monitoring possibilities. This may ultimately lead to a user-modifiable technology that could be tailored to various patients, disease modalities, or hospital settings.

Another area of improvement that was recognized is the perceived lack of precision in information presentation. Hereby, a subgroup of participants expressed a desire for more gradient differentiation of certain parameters, such as the color gradient of the oxygen saturation display (Participants 17, 18, and 21). While previous qualitative studies have highlighted this particular theme [35], with users suggesting that a higher level of differentiation would provide more visual information and a more complete representation of patient status, the Visual Patient Avatar is not intended to replace conventional patient monitoring as a quantification tool. Instead, the technology is designed to complement traditional vital-sign displays by providing a prompt initial evaluation of the patient status and to quickly guide the caregiver to the correct conventionally displayed parameter that requires attention. Additionally, it is important to consider that, in a previous study, a simplified gradient showed a greater interrater reliability and thus provided optimized situation awareness for care providers [22]. Therefore, to ensure clarity and simplicity, most parameters are visualized along a four-step gradient, indicating whether the parameter in question is undetected, below, within, or above a safe range.

Finally, some participants suggested that there may be room for improvement in space allocation and the display of the Visual Patient Avatar when multiple parameters are out of the normal range. The University Hospital Zurich currently offers a modifiable size setting for the visualization technology, which is set based on institutional standards. The size setting of the Visual Patient Avatar should be considered on an individual basis, with the introduction of the technology to new institutes. Regarding the issue of multiple parameters being outside of the normal range, an increase in complexity is to be expected to a certain degree, as care providers must interpret several less frequently occurring states simultaneously. We expect that, with further use of the technology, the interpretation complexity will reduce, as users will naturally adjust to these states. In the ongoing development of the Visual Patient Avatar, it is important to consider these inputs and implement improvements that do not compromise situation awareness.

It is worth noting that the introduction of novel tools requires a period of user adaptation, which should ultimately lead to a cultural shift in the work environment towards acceptance of a new technology. This study assessed the initial perceptions of healthcare professionals regarding the first-time implementation of the Visual Patient Avatar in the adult anesthesia setting and allowed for the collection of valuable insights for further development.

This study exhibits several limitations. First, this study is constrained by the inherent limitations associated with qualitative research. The conclusions drawn from qualitative analysis cannot be extrapolated to broader populations with the same degree of confidence as quantitative outcomes, as they are not subjected to statistical significance testing [36].

Moreover, this study was conducted within a single European university hospital, the initial development site for the technology. Consequently, user perspectives could exhibit variability across diverse global clinical contexts. Also, this study focused on medical staff’s first impressions of the technology introduced three months previously in the OR. As with most novel technologies, users require a period of adaptation to become accustomed to and accept a new tool, which may cause their perceptions to change over time. Therefore, the need for further qualitative and quantitative research persists.

Finally, despite our measures to address these effects, the outcomes of the survey may be subject to nonresponse, self-selection, and information bias [26,27,28]. Nonresponse may lead to incomplete and biased data if certain individuals opt not to participate, affecting the accuracy and generalizability of findings. Given the demanding clinical responsibilities of our pool of participants, who are practicing clinicians, time constraints may have played a role in nonresponse. Regarding the concern of self-selection bias, a sample comprising individuals participating based on their personal interests, experiences, or motivations may not be entirely representative of the target population as a whole. Given the early evaluation of the tool at its place of origin, where an inherent emotional connection exists, it is reasonable to anticipate a degree of the described self-selection bias, as individuals with stronger opinions about the newly introduced technology may be more likely to participate in the study. Also, although we have taken measures to develop a valid data collection instrument to maximize generalizability, information bias may still be a factor affecting survey results.

The study’s primary strength lies in its novelty as the inaugural investigation into user perceptions of the Visual Patient Avatar technology in adult anesthesia following its implementation in clinical practice. The qualitative assessment of very early user perceptions of the tool provides a valuable strategic roadmap for the technology’s further development.

## 5. Conclusions

This study is the first to explore the perspectives of anesthesia providers working with adult patients regarding the Philips Visual Patient Avatar technology after its integration into clinical practice. It reveals that anesthesia providers generally find the visual data presented by Visual Patient Avatar to be a valuable complement to conventional displays, as such data offer a swift synopsis of the clinical situation and facilitate the prompt identification of issues. The assessment of user perceptions of the novel technology also identified certain areas for improvement suggested by the handling clinicians. For instance, an important user desire for enhancement was the wish for adjustable Visual Patient Avatar thresholds. Potential solutions suggested by study participants, such as refinements to visualization designs and user-customizable thresholds or thresholds that align with predefined audible alarm limits, warrant further research and development.

## Figures and Tables

**Figure 1 bioengineering-11-00323-f001:**
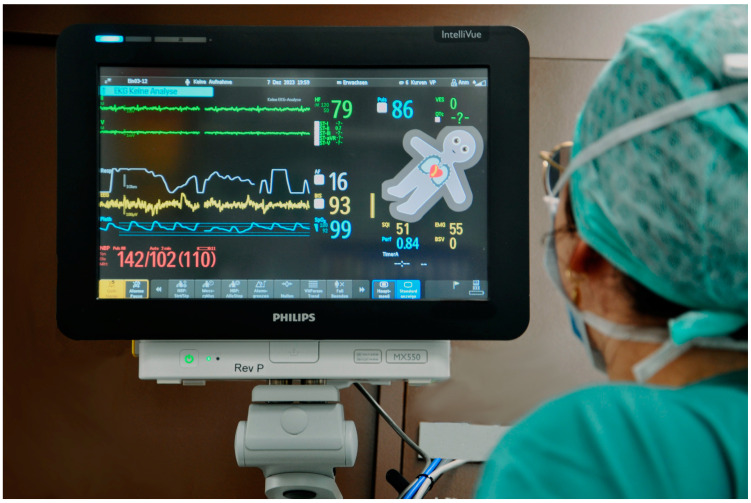
Philips Visual Patient Avatar in the operating room setting, displayed as a split-screen feature alongside conventional vital-sign parameter monitoring.

**Figure 2 bioengineering-11-00323-f002:**
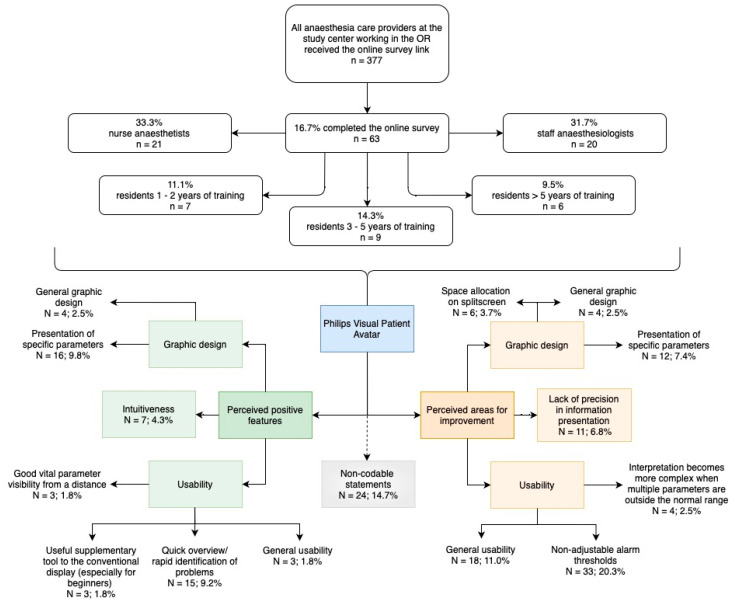
Flowchart illustrating the response rate and distribution of participants (n), breakdown of comments describing positive features and areas for improvement, and themes identified, including the number and percentage of statements (N).

**Figure 3 bioengineering-11-00323-f003:**
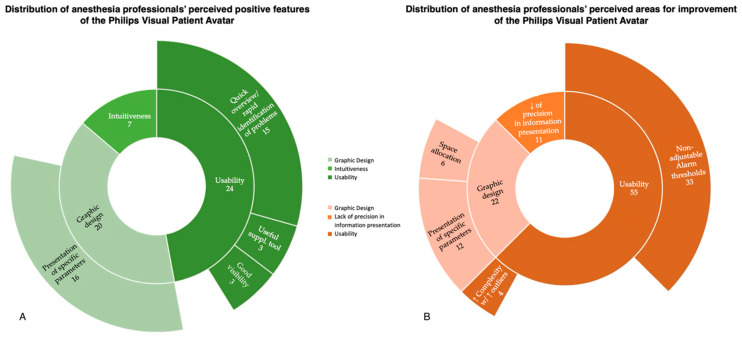
Sunburst diagrams representing the hierarchical distribution of anesthesia professionals’ perceptions of the Philips Visual Patient Avatar. (**A**) Perceived positive features. (**B**) Perceived areas for improvement. ↑: increased. ↓: decreased.

**Table 1 bioengineering-11-00323-t001:** Participant characteristics.

Participants (*n* = 63)		
Sex	Female	31 (49.2%)
	Male	29 (46.0%)
	Other gender identity	3 (4.8%)
Participant age in years	20–35	23 (36.5%)
	36–50	32 (50.8%)
	51–65	8 (12.7%)
Work in anesthesia experience in years, median (IQR)		8 (4–13.5)
Role	Nurse anesthetist in training	1 (1.6%)
	Certified nurse anesthetist	20 (31.7%)
	Resident 1–2 years of training	7 (11.1%)
	Resident 3–5 years of training	9 (14.3%)
	Resident with >5 years of experience	6 (9.5%)
	Staff anesthesiologist	20 (31.7%)
Educational material consulted	Short manual only	17 (27.0%)
	Tutorial video only	7 (11.1%)
	Both	33 (52.4%)
	None	6 (9.5%)

**Table 2 bioengineering-11-00323-t002:** Perceived positive features and areas for improvement presented alongside the corresponding number of statements, percentages, and examples.

**Perceived Positive Features**	
Usability (24/163; 14.6%)	
Quick overview/rapid identification of problems (15/163; 9.2%)	*Quick overview of various parameters.* (Participant 37) *Problems are quickly and easily identified.* (Participant 15)
Good vital parameters visibility from a distance (3/163; 1.8%)	*Ability to assess the situation, even from a* *distance, when numbers are no longer legible.* (Participant 7) *Good visibility of alarms, even from a* *greater distance.* (Participant 9)
Useful supplementary tool to the conventional display (3/163; 1.8%)	*Useful supplementary tool.* (Participant 25) *A good addition to the numerical display.* (Participant 19)
Graphic design (20/163; 12.3%)	
Presentation of specific parameters (16/163; 9.8%)	*The purple color immediately catches the eye, indicating a drop in saturation.* (Participant 54) *The different colors and symbols are easily distinguishable and provide good indicators.* (Participant 61)
Intuitiveness (7/163; 4.3%)	
*You get used to it very quickly.* (Participant 42) *I like the clarity provided by Visual Patient.* (Participant 7)	
**Perceived Areas for Improvement**	
Usability (55/163; 33.7%)	
Non-adjustability of alarm thresholds (33/163; 20.3%)	*User-set alarm thresholds should be automatically adopted.* (Participant 3) *For SpO2, you may want to enter changeable values so that a COPD patient is not displayed as purple all the time.* (Participant 14)
Interpretation becomes more complex when multiple parameters are outside of the normal range (4/163; 2.5%)	*It is difficult to perceive the information when multiple problems occur simultaneously.* (Participant 34) *In general, are there already too many parameters displayed in Visual Patient and should it be reduced to the basics?* (Participant 47)
Graphic design (22/163; 13.5%)	
Presentation of specific parameters (12/163; 7.4%)	*I would like the inclusion of pacemaker function (as in a regular EKG).* (Participant 38) *The pulsation of Visual Patient indicates whether the pulse is within the normal range. However, the frequency of the pulsation does not match the heartbeat heard* via *SpO2. This is very confusing because what is seen does not correspond to what is heard.* (Participant 37)
Space allocation on split screen (6/163; 3.7%)	*Too many pieces of information on too little space.* (Participant 44) *The reduction of the remaining curves is a major point of criticism.* (Participant 9)
Lack of precision in information presentation (11/163; 6.8%)	
*The cyanosis could be further differentiated in color.* (Participant 18) *It would be helpful if the color could be further differentiated. Similarly, CO*_2_*could be color-coded for hyper- or hypocapnia.* (Participant 21)	

## Data Availability

The datasets used and/or analyzed during the current study are available from the corresponding author upon reasonable request.

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
