# Peer review of "Qualitative Exploration of Anesthesia Providers’ Perceptions Regarding Philips Visual Patient Avatar in Clinical Practice"

_bioengineering, 2024, doi:10.3390/bioengineering11040323_

Round 1
Reviewer 1 Report
Comments and Suggestions for Authors
The novelty of the study should be explained.
The results of the study should be presented by graphs and diagrams.
Comments on the Quality of English Language
The English language should be modified.
Reviewer 2 Report
Comments and Suggestions for Authors
1. The authors should create a table for Figure 1 to compare and discuss the differences from traditional vital sign parameter monitoring.
2. The authors should utilize multiple translation software programs to discuss and analyze their data.
3. Whether the participants selected by the authors are random and whether there is a bias in their age and gender.
4. At present, deep learning technology has strong analytical and processing capabilities for many medical data, and the authors should consider using stack autoencoder networks.
5. In disease diagnosis, hyperspectral imaging systems also have important applications, such as literature “A stare-down video-rate high-throughput hyperspectral imaging system and its applications in biological sample sensing”, which is an effective supplement to many current imaging methods. It is recommended that the author elaborate on the application of hyperspectral imaging in disease diagnosis.
Reviewer 3 Report
Comments and Suggestions for Authors
As the authors rightly point out, this study has the limitations of a qualitative study; however, it deals with a very interesting topic and, despite its limitations, it is well designed and well conducted and in the analysis of the results it also takes into account the mental processes of adaptation and acceptance of technological innovations that can represent a barrier in healthcare contexts.
It might be interesting to re-evaluate the questionnaire covered by the survey after a more prolonged period (e.g. 6-12 months) of use of the Philips Visual Patient Avatar, perhaps after the implementation of an updated version of the technology that incorporates some of the suggestions received from anesthesia professionals who participated in the current study.
It would also be appropriate to repeat the survey in other centres, not emotionally linked to the technology under examination, i.e. not developers of the aforementioned technology.

Round 2
Reviewer 1 Report
Comments and Suggestions for Authors
The paper can be accepted.
Reviewer 2 Report
Comments and Suggestions for Authors
Accept in present form